# Epidemiology and severity risk factors of dengue virus infection during the 2023-2024 outbreak in Colombia

Daniela Torres-Hernández[1], Nathan D. Grubaugh[2,3], Mónica A. Murillo-Ortiz[4,5], Isabel C. Hurtado[1,4,6], Verity Hill[3], Mallery I. Breban[3], Mara Gómez-Zambrano[1], Pio López[1,9], Erika Cantor[7], Diana M. Dávalos[9], Eduardo López-Medina [1,8,9]*

1 Department of Pediatrics, Universidad del Valle, Cali, Colombia, 2 Department of Ecology and Evolutionary Biology, Yale University, New Haven, Connecticut, United States of America, 3 Department of Epidemiology of Microbial Diseases, Yale School of Public Health, New Haven, Connecticut, United States of America, 4 Hospital Universitario del Valle. Cali, Colombia, 5 School of Bacteriology, Universidad del Valle, Cali, Colombia, 6 State Department of Public Health, Valle del Cauca, Colombia, 7 Department of Clinical Epidemiology and Biostatistics, Pontificia Universidad Javeriana, Bogotá, Colombia, 8 Clínica Imbanaco Grupo Quironsalud, Cali, Colombia, 9 Department of Scientific Advancement, EndPoints Network of Research Sites, Latin America

* eduardo.lopez@ceiponline.org

## Abstract

### Introduction

During 2023–2024, the Americas faced its largest dengue epidemic to date. We used a detailed dengue classification to identify patients with serious manifestations of dengue and aimed to describe risk factors for occurrence.

### Methods

From April 2023 to September 2024, we conducted a prospective in-hospital active case-finding cohort study at the Hospital Universitario del Valle (HUV) in Colombia, enrolling patients of all ages with virologically confirmed dengue (VCD). Sociodemographic and clinical data were collected, and the dengue virus (DENV) genome was sequenced for serotyping and genotyping. Multivariable logistic regression modeling identified factors associated with "serious manifestations of dengue", defined as severe dengue (2009 WHO classification) or dengue with warning signs (DwWS) with a more severe course, including vascular leakage (increases in hematocrit >20%, pleural effusion or ascites with hemodynamic/respiratory compromise); bleeding with hemodynamic instability or requiring blood transfusion; thrombocytopenia <20,000 or organ dysfunction (myocarditis, encephalitis or liver failure).

### Results

Among 600 patients (median age 13 years, 55% male), serotyping and genotyping were possible for 340 (57%) and 296 (49%) samples, respectively. The

**Data availability statement:** DENV genomes are available at National Center for Biotechnology Information BioProject (https://www.ncbi.nlm.nih.gov/bioproject; accession no. PRJNA1132139). All other relevant data are in the manuscript and its supporting information files.

**Funding:** This study was supported by the National Institute of Allergy and Infectious Diseases of the National Institutes of Health (NIH) under award no. DP2AI176740 to NDG and by an NIH Shared Equipment grant (no. 1S10OD028669-01) awarded to NDG at Yale Center for Genome Analysis. Additional support was received through an Investigator-Initiated Research Agreement (IISR-2023-200390) to ELM from Takeda. The funders had no role in study design, data collection and analysis, decision to publish, or preparation of the manuscript.

**Competing interests:** I have read the journal's policy and the authors of this manuscript have the following competing interests: ELM and/or his institution have received research grants from Takeda, MSD, Sanofi Pasteur, Janssen, and GSK, and speaker honoraria from Takeda, MSD, Sanofi Pasteur, and GSK.

most frequent serotypes were DENV-2 (32%) and DENV-3 (15%), and lineages 2II_F.1.1.2 (19%) and 3III_C.1 (15%). Serious manifestation of dengue occurred in 167 (28%) patients, including 22 with severe dengue and 145 DwWS cases with a more severe course. Independent risk factors for serious manifestations of dengue were living outside of city limits, presenting with edema and higher leukocyte counts, whereas lower odds were observed for patients with higher platelets and lymphocyte counts, and infections other than DENV-2. History of dengue infection showed no significant effect on the risk of serious clinical manifestations across the different serotypes.

## Conclusion

This outbreak involved multiple dengue virus serotypes and genotypes and predominantly affected children and adolescents. Identification of the risk factors described here could enable earlier recognition of patients with serious dengue manifestations. In this cohort, serotype 2 was associated with higher risk but given the unpredictable dynamics of severe dengue, efforts should strive for tetravalent protection, regardless of prior dengue exposure.

## Author summary

Dengue remains a serious public health concern, especially during major outbreaks like the 2023–2024 epidemic in Colombia. This study focuses on patients with "serious manifestations of dengue", a commonly seen yet insufficiently studied subgroup frequently requiring intensive medical care. By analyzing sociodemographic, clinical, and viral factors in patients presenting to a referral hospital during the outbreak, this study identifies possible predictors of progression to serious disease. The study shows that a high frequency of patients with dengue with warning signs develop "serious manifestations", especially those living outside of the state's capital, and those presenting with edema, increased leukocytes and lower platelets and lymphocytes count. Different serotypes and lineages co-circulated; infections with DENV-2 were associated with a higher risk of severe disease. Previous dengue exposure did not affect the risk of serious manifestations, highlighting the need for safe and effective vaccines for people living in endemic regions, whether or not they have had documented dengue in the past.

This work emphasizes the complexity of dengue pathogenesis, driven by viral genotype, immune history, and host factors, and provides critical insights for early risk identification and tailored interventions to reduce dengue-related morbidity and mortality.

## Introduction

The 2024 dengue epidemic in the Americas is considered the worst in the documented history of the region, with over 13 million annual cases, including more than 8000 deaths [1]. Models using different statistical techniques estimate that case numbers will continue to increase in various world regions in the coming years unless effective measures are widely implemented [2,3].

Although most dengue virus (DENV) infections are asymptomatic, a small subgroup of patients may progress to severe dengue [4]. In recent years, there has been increasing recognition that severe dengue is a rare event, occurring in less than 1% of dengue cases [5], and that less severe manifestations that require early recognition and medical interventions significantly contribute to the overall disease burden [4,6]. In addition, dengue with warning signs (DwWS) or severe dengue according to the World Health Organization (WHO) 2009 classification is of limited use in clinical research because diagnostic criteria are not well-defined and may lead to misclassification [6,7]. 2009 WHO guidelines recommend that all patients with warning signs be admitted to a hospital [8], which, during epidemics, may result in over-admission of patients with subjective findings according to the treating physician (i.e., abdominal pain) but without any objective evidence of clinical significance. Therefore, an expert working group convened by the National Institute of Allergy and Infectious Diseases (NIAID), the National Institutes of Health, and the Partnership for Dengue Control (PDC) has developed candidate, detailed definitions for moderate and severe disease that provide well-defined criteria for a subgroup of patients, especially those with DwWS who have more serious clinical courses [6].

While previous studies have evaluated the risk factors for severe dengue or dengue with warning signs as per the 2009 WHO classification [9,10], no studies have investigated the factors associated with the most serious disease spectrum in patients with DwWS. These patients are frequently encountered and present with serious manifestations that require early identification, careful monitoring, and frequent intensive medical support [11]. Thus, they constitute a recently identified and critical subgroup within the broader population of dengue patients, designated as "serious manifestations of dengue." The purpose of this study was to evaluate the sociodemographic, viral, and clinical risk factors associated with serious manifestations of dengue in a population presenting to the emergency department of a referral hospital during the large 2023–24 dengue outbreak in Valle state, Colombia.

## Methods

### Ethics statement

The study was approved by the Ethical Committee of the Hospital Universitario del Valle (approval no. HUV-022–2023) and the Corporación Científica Pediátrica independent Ethics Committee (approval no. CEI-1548–2022). Written informed assent or consent was obtained from all participants or their parents or legal guardians before enrollment. Molecular analysis of collected blood samples was approved by the Yale University Human Research Protection Program (protocol no. 2000033281).

### Study design, participants and setting

This is a prospective in-hospital active case finding cohort study of all patients who received care due to virologically confirmed dengue (VCD) at Hospital Universitario del Valle (HUV) in the city of Cali during the 2023–2024 dengue season, a time period during which no dengue vaccines were available (15 April 2023 until 30 Aug 2024). HUV is a reference, government-funded level 3 hospital where patients from Cali and surrounding municipalities and states receive advanced care. Cali is the state's capital and the third largest city in Colombia with an estimated population of 2.3 million living within city limits. This study was approved by the independent ethics committee of Corporación Científica Pediátrica and HUV (approval no. CEI-1548–2022 & HUV-022–2023) and was conducted in accordance with Good Clinical Practice and the guidelines of the Declaration of Helsinki. Written informed assent or consent was obtained from all participants or their

parents or legal guardians before enrollment. Molecular analysis of collected blood samples was approved by the Yale University Human Research Protection Program (protocol no. 2000033281).

Active case finding was performed daily by study staff who visited the microbiology laboratory to identify patients with VCD and enroll them in the study. In addition, considering that patients with VCD may have been transferred from different institutions and therefore not tested at HUV, surveillance was also performed at the different HUV wards where patients with dengue might have been receiving care.

Suspected dengue was defined as episodes in which a diagnostic dengue test was performed at HUV based on clinical suspicion. VCD was diagnosed in patients with a documented fever (>38 °C) of less than 7 days duration and one of the following manifestations: Headache, retroocular pain, myalgia, arthralgia, nausea, vomiting or rash [12], along with a positive NS1 for patients with symptom onset of 5 days or less, or IgM dengue enzyme-linked immunosorbent assay (ELISA) for patients with symptom onset of 6 days or more. DENV evaluation was performed with the VIDAS dengue panel (bioMérieux, Marcy-l'Étoile, France).

In addition, retrospective data on suspected dengue and VCD cases reported by HUV to the State Health Department, and VCD cases reported from the State Health Department to the national surveillance system were collected from the period January 2021 – 2024. Data were obtained from the hospital's laboratory database and statistics office, and from the national surveillance system in public health, SIVIGILA.

## Variables and data sources

For patients with VCD who signed an informed consent (or informed assent for minors with parental consent), data were collected via interview or review of the medical record and patients were followed by study staff until discharge. A blood sample was collected within the first 7 days of symptoms for serotyping and lineage identification via PCR and sequencing. For patients who were not hospitalized, a follow-up phone call was made approximately 1 week later to assess for definitive outcomes.

For each collected serum sample, 140 µL was used for viral RNA extractions using the QIAamp Viral RNA Mini Kit (QIAGEN Inc., Germantown, MD, USA.) according to manufacturer's instructions. Then, identification of DENV serotypes (DENV-1 to -4) was performed on all samples using the CDC DENV-1–4 rRT-PCR Multiplex Assay for DENV typing [13] and additionally from patients with VCD using the VIASURE Dengue Serotyping Real Time PCR Detection Kit from CerTest Biotec. Whole-genome sequencing was performed using DengueSeq. [14] Bioinformatics analysis, including primer trimming and consensus generation, was conducted with the iVar pipeline [14]. Samples with ≥5% genome completeness were assigned DENV lineages using Genome Detective and Nextclade [15]. The lineage classifications were used to verify the PCR-based serotype calls. DENV genomes greater than 70% completeness are available at National Center for Biotechnology Information BioProject (https://www.ncbi.nlm.nih.gov/bioproject; accession no. PRJNA1132139).

Patients were assessed and managed in accordance with the guidelines established by the WHO [8], and the clinical judgment of their attending physician. The exposure variables included sociodemographic factors, laboratory results, clinical characteristics, viral parameters (including serotype and lineage), comorbid conditions, and the duration of fever from onset to the time of seeking medical care. History of DENV infection was defined as a positive IgG in the VIDAS dengue panel obtained in the first 7 days of symptoms.

The primary outcome variable was the occurrence of serious manifestations of dengue. The serious manifestations of dengue classification was defined as severe dengue or DwWS, both 2009 WHO classifications, that progressed to a more severe disease course [6]. This included any of the following: 1. Severe plasma leakage (hemodynamic instability or respiratory compromise, and evidence of plasma leakage defined by: increases in hematocrit >20%; and/or new pleural effusion or ascites). 2. Severe bleeding with hemodynamic instability or requiring blood transfusion. 3. Severe thrombocytopenia (<20,000 platelets/uL). 4. Organ dysfunction (myocarditis, encephalitis or liver failure).

**Neglected Tropical Diseases** PLOS

A sensitivity analysis was used to identify factors associated with severe dengue according to the 2009 WHO classification. For this analysis, multivariable analysis was not carried out due to the low case numbers of severe dengue.

## Analysis

Quantitative variables were summarized using medians and interquartile ranges, while qualitative variables were described using frequencies and percentages. A univariate logistic regression model was estimated using serious manifestations of dengue as the dependent variable. A multiple logistic regression model was fitted, incorporating all independent variables that had a p-value <0.25 in the univariate analysis, without applying a variable selection algorithm. Variable selection was avoided because the number of events per variable (EPV) was < 10 and in such cases variable selection is not recommended [16].

Multicollinearity was assessed using the variance inflation factor. During the development of the multiple logistic regression model, an interaction term between the infecting serotype and previous dengue exposure status was evaluated. P values <0.05 were considered statistically significant. The average number of VCD cases reported by HUV to the State Health Department, and VCD cases reported from the State Health Department to the national surveillance system during the epidemic period (April 2023–2024) was compared to that of the 2021–March 2023 period. All analyses were performed using R (version 4.4.1, R Core Team).

## Results

The 2023–2024 dengue outbreak was the worst on record in Valle state, Colombia [17]. Compared to the 2021-March 2023 period, the number of VCD cases reported by the State Health Department and HUV increased more than 400% (Fig 1).

During this study period, there were 1403 suspected dengue cases at HUV, 600 of whom had VCD and were included in this study. Using the 2009 WHO classification, there were 35 cases of dengue without warning signs (DwoWS), 543 DwWS and 22 severe dengue cases. Using a more granular classification of the DwWS [6], 145 had a critical disease course, 254 had moderate disease and 144 had mild disease. Therefore, there were 167 cases of serious manifestations of dengue (22 severe + 145 DwWS critical disease course) and 433 cases of non-serious manifestations of dengue (35 DwoWS + 144 DwWS mild disease course + 254 DwWS moderate disease course) (Fig 2).

Median patient age was 13 years (interquartile range 8–19), and most were male, students, of mixed race and living in the city of Cali. Table 1 presents the demographic, clinical and viral characteristics of these patients according to their disease severity.

In the bi-variate analysis, presenting signs or symptoms and laboratory findings differed between patients with serious and non-serious manifestations of dengue (Table 1). Patients who developed serious manifestations of dengue more frequently presented with diarrhea, bleeding and edema, and had higher leukocytes and lower platelet counts. A prior history of DENV infection, indicated by a positive IgG serology, was also more commonly observed in patients who developed serious manifestations of dengue [OR 1.86 (1.24, 2.81) p-value 0.003], as well as having an infection by lineages DENV-2II-F.1.1.2 or 2III_D.2. DENV-1 and DENV-3 had lower risk as compared to DENV-2, although the serotype was unknown in 43% of the samples due to virus titters below our PCR and sequencing limits.

While all four DENV serotypes were detected during the study period, serotypes 1, 2 and 3 co-circulated in most months and DENV-2 was the dominant serotype, with DENV-4 appearing only sporadically (Fig 3a). Several DENV lineages were detected (Table 1) and circulated simultaneously (Fig 3b). During the peak of the outbreak, the emerging lineage DENV-2II_F.1.1.2 mostly replaced DENV-2III_D.2 while becoming the dominant lineage among all serotypes during the rest of the outbreak [18].

Independent risk factors for serious manifestations among patients with dengue were identified using multivariable logistic regression models (Table 2). Contracting dengue outside of Cali (the state capital), edema and higher leukocyte

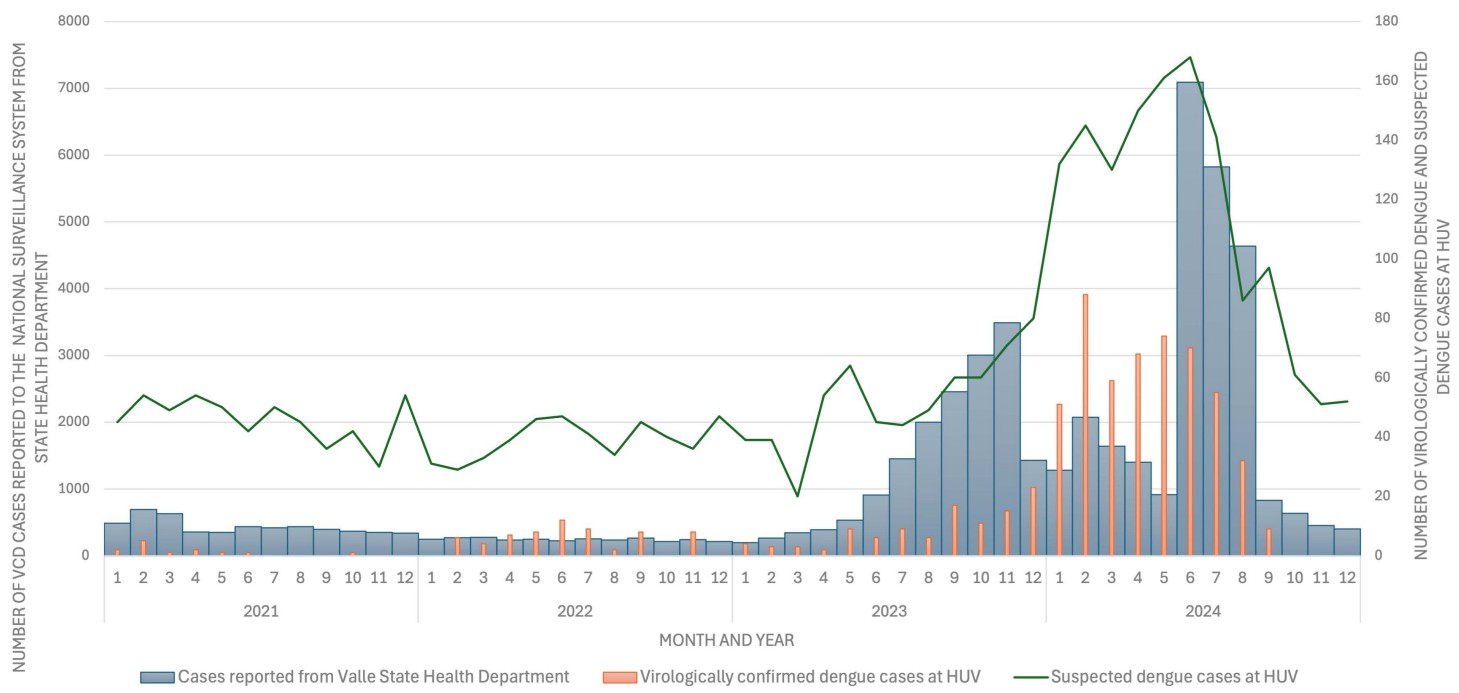

**Fig 1. Evolution in the number of cases of VCD reported to the National Surveillance System from State Health Department and VCD and suspected dengue cases at HUV by month and year, 2021-2024.**

counts were associated with serious manifestations, whereas lower odds were observed for higher platelets and lymphocyte counts. Infections by any serotype had lower odds of severe manifestations compared with DENV-2; for DENV-1, the odds ratio was 0.56 (95% CI 0.24–1.24, p = 0.16), indicating uncertainty in its effect.

In the interaction model, after adjusting for age, sex, city of origin, education level, and occupation, history of dengue infection (IgG-positive status) showed no significant effect on the risk of serious clinical manifestations across the different serotypes (likelihood-ratio test: p = 0.38).

Patients with serious manifestations of dengue had poorer outcomes than patients with non-serious manifestations. Overall, 104 patients (17%) required ICU admission, especially patients with serious manifestations (38% vs. 9%, P=<0.001). Eleven patients required vasopressor medications, all of them with serious manifestations (Table 3).

Sensitivity analysis revealed similar associations between exposure variables and severe dengue according to the 2009 WHO classification; however, AST levels were elevated in patients with severe dengue, and no significant association was observed with DENV-2. The limited number of severe cases precluded further analysis (S1 Table).

## Discussion

This study, conducted during the 2023–2024 dengue outbreak, documented the largest recorded epidemic in Colombia to date and provides critical insights into the sociodemographic, viral, and clinical risk factors associated with serious manifestations of dengue. With 320,982 reported cases nationwide, including 230,746 confirmed cases [19], the scale of this outbreak surpassed prior epidemics, such as those in 2019–2020 (127,553 cases) and the historic 2010 peak (157,203 cases) [17]. In the context of this significant large-scale outbreak, we identified variables to facilitate early detection of serious manifestations and enable timely interventions that can mitigate life-threatening complications or death.

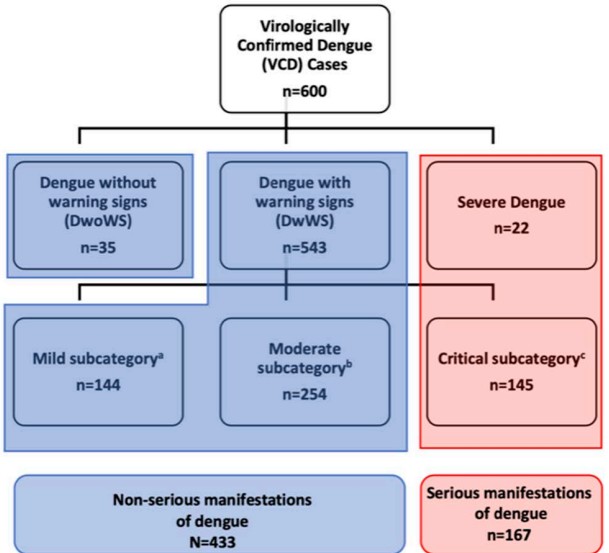

**Fig 2. Flowchart depicting the classification of Serious and Non-serious dengue manifestations using a granular approach to the dengue without warning signs (DwoWS) category. ª.** Patients with mild dengue do not exhibit vascular leakage (defined as an increase in hematocrit of at least 15% during the illness, pleural effusion or ascites), have no bleeding or minor bleeding that do not require local intervention, have platelets > 50,000/mm³, and no evidence of shock, hemodynamic instability, or organ dysfunction. **ᵇ.** Patients with moderate dengue exhibit no hemodynamic instability or respiratory compromise but show *Plasma Leakage:* ≥ 15% change in hematocrit during illness; New pleural effusion, pericardial effusion, or ascites on imaging. *Bleeding (without shock or need for transfusion but requiring local intervention):* Large skin/injection site bleeds needing compression; Nose/gum bleeds requiring intervention (e.g., packing, adrenaline); Gastrointestinal/vaginal bleeding requiring monitoring and type/crossmatch; Persistent bleeding despite local measures, needing intensive monitoring. *Platelet Count:* Between 20,000 – 50,000/mm³. *Liver Involvement:* Acute viral hepatitis symptoms; ALT ≥ 400 U/L; No signs of acute liver failure (no mental status changes, INR < 1.5). *Myocarditis:* (one of the following): Elevated troponin, CPK-MB or ST2 above the laboratory upper limits of normal; New onset cardiac arrhythmia or abnormal ECG. *Neurologic Disease (all must be met):* Glasgow Coma Score 12-14 for <2 days; No need for intensive interventions (intubation, shunting, ICU); No lasting impairment beyond 48 hours. **ᶜ.** Patients with critical dengue exhibit hemodynamic instability or respiratory compromise, and evidence of *Severe Plasma Leakage:* > 20% hematocrit change during illness; new pleural effusion, pericardial effusion, or ascites on imaging. *Severe Bleeding (any of the following):* Bleeding into a critical organ (e.g., CNS bleed); Bleeding causing hemodynamic instability; Bleeding leading to death or permanent disability (e.g., CNS/intraocular bleed); Bleeding requiring blood transfusion and intensive care monitoring. *Severe Thrombocytopenia:* Platelet count <20,000/mm³, requiring intensive observation or ICU transfer. *Liver Failure (all must be met):* Clinical acute hepatitis; New-onset mental status changes or hepatic encephalopathy and Coagulopathy (INR ≥ 1.5). *Myocarditis (criteria 1 or 2, plus criteria 3):* 1). Acute illness with discrete onset of signs and symptoms consistent with acute viral myocarditis (e.g., elevated troponin, CPK-MB, or ST2 above the laboratory upper limit of normal); 2). New-onset cardiac arrhythmia or ECG abnormalities. 3). Need for inotropic support and myocardial dysfunction on echocardiogram. *Neurologic Disease (all must be met):* Glasgow Coma Score <11 (or equivalent pediatric scores). Neurologic complications leading to death, disability. intubation, or ICU care.

While the reasons for the recent outbreak's large magnitude are unclear, investigations have increasingly examined the impact of genetic diversity among DENV serotypes [20,21] and genotypes [22–24] on the rising incidence of severe dengue cases [25]. The concurrent circulation of multiple DENV serotypes and lineages identified in this study may have contributed to increased case numbers and greater severity due to sequential infections [26–29] and variable phenotypes that affect virulence, transmissibility, and immune evasion [18]. This phenomenon, in part due to serotype reintroductions or replacements, could partially explain the rising severity of recent outbreaks.

Previous studies, including a systematic review from different Latin American countries, have identified risk factors for severe dengue based on the 2009 WHO classification [10,26]. Severe dengue is a condition that occurs in less than 1% of all cases and typically manifests in the later stages of the disease [3,12]. However, patients with DwWS may experience complicated clinical courses requiring intensive care and invasive interventions, despite not meeting the criteria for severe dengue [11]. Our study's classification of serious manifestations of dengue captures both severe dengue and cases with

**Table 1. Sociodemographic and clinical characteristics of patients with serious and non-serious manifestations of dengue.**

| Variable | Non-serious manifestation of dengue N=433 | Serious manifestations of dengue N=167 | Total N=600 | OR [95 CI] | p- value |
|---|---|---|---|---|---|
| Age of patient, median years (IQR) | 12.0 (8.0 to 17.0) | 16.0 (9.5 to 29.5) | 13.0 (8.0 to 19.0) | 1.02 [1.01, 1.03] | <0.001 |
| **Sex n(%)** | | | | | |
| Female | 193 (44.6) | 79 (47.3) | 272 (45.3) | – | |
| Male | 240 (55.4) | 88 (52.7) | 328 (54.7) | 0.9 [0.63, 1.28] | 0.547 |
| **Race n(%)** | | | | | |
| White | 66 (15.2) | 26 (15.6) | 92 (15.3) | – | |
| Black | 33 (7.6) | 16 (9.6) | 49 (8.2) | 1.23 [0.57, 2.59] | 0.587 |
| Mix race | 334 (77.1) | 125 (74.9) | 459 (76.5) | 0.95 [0.58, 1.59] | 0.840 |
| **Level of education[a] n(%)** | | | | | |
| Less than Basic | 205 (47.3) | 68 (40.7) | 273 (45.5) | – | |
| Basic/intermediate | 183 (42.3) | 72 (43.1) | 255 (42.5) | 1.19 [0.81, 1.75] | 0.387 |
| Advanced | 45 (10.4) | 27 (16.2) | 72 (12.0) | 1.81 [1.04, 3.12] | 0.035 |
| **Place of origin n(%)** | | | | | |
| Urban | 396 (91.5) | 153 (91.6) | 549 (91.5) | – | |
| Rural | 37 (8.5) | 14 (8.4) | 51 (8.5) | 0.98 [0.50, 1.82] | 0.949 |
| **City of origin n(%)** | | | | | |
| Cali | 283 (65.4) | 79 (47.3) | 362 (60.3) | – | |
| Other | 150 (34.6) | 88 (52.7) | 238 (39.7) | 2.1 [1.46, 3.02] | <0.001 |
| **Occupation n(%)** | | | | | |
| House wive | 25 (5.8) | 18 (10.8) | 43 (7.2) | – | |
| Employee | 45 (10.4) | 22 (13.2) | 67 (11.2) | 0.68 [0.31, 1.50] | 0.338 |
| Freelance work | 8 (1.8) | 7 (4.2) | 15 (2.5) | 1.22 [0.37, 4.00] | 0.746 |
| Unemployed/none | 60 (13.9) | 31 (18.6) | 91 (15.2) | 0.72 [0.34, 1.52] | 0.383 |
| Student | 295 (68.1) | 89 (53.3) | 384 (64.0) | 0.42 [0.22, 0.81] | 0.009 |
| **Body mass index[b] n(%)** | | | | | |
| Underweight | 27 (6.2) | 13 (7.8) | 40 (6.7) | – | – |
| Normal | 180 (41.6) | 61 (36.5) | 241 (40.2) | 1.42 [0.67, 2.88] | 0.341 |
| Overweight | 91 (21.0) | 37 (22.2) | 128 (21.3) | 1.2 [0.74, 1.93] | 0.457 |
| Obese | 135 (31.2) | 56 (33.5) | 191 (31.8) | 1.22 [0.80, 1.87] | 0.352 |
| **Any comorbidity n(%)** | 60 (13.9) | 27 (16.2) | 87 (14.5) | 1.20 [0.72, 1.95] | 0.472 |
| Pulmonary disease | 12 (2.8) | 4 (2.4) | 16 (2.7) | | |
| Cardiovascular disease | 8 (1.8) | 8 (4.8) | 16 (2.7) | | |
| Neurological disease | 15 (3.5) | 4 (2.4) | 19 (3.2) | | |
| Metabolic disease | 11 (2.5) | 4 (2.5) | 15 (2.5) | | |
| Kidney disease | 6 (1.4) | 3 (1.8) | 9 (1.5) | | |
| Immunosuppression | 7 (1.6) | 4 (2.4) | 11 (1.8) | | |
| Hematological disease | 1 (0.2) | 0 (0.0) | 1 (0.2) | | |
| None | 373 (86.1) | 140 (83.8) | 513 (85.5) | | |
| Time fever to consult (days), median (IQR) | 3.0 (1.0 to 4.0) | 2.0 (1.0 to 4.0) | 3.0 (1.0 to 4.0) | 0.96 [0.86, 1.07] | 0.511 |
| ***Clinical symptoms* n(%)** | | | | | |
| Headache | 380 (87.8) | 150 (89.8) | 530 (88.3) | 1.23 [0.70, 2.25] | 0.482 |
| Retro-ocular pain | 291 (67.2) | 120 (71.9) | 411 (68.5) | 1.25 [0.85, 1.86] | 0.272 |
| Mialgias/ Arthralgias | 384 (88.7) | 156 (93.4) | 540 (90.0) | 1.80 (0.95, 3.75) | 0.087 |

*(Continued)*

**Table 1.** (Continued)

| Variable | Non-serious manifestation of dengue N=433 | Serious manifestations of dengue N=167 | Total N=600 | OR [95 CI] | p- value |
|---|---|---|---|---|---|
| Rash | 236 (54.5) | 89 (53.3) | 325 (54.2) | 0.95 [0.67, 1.36] | 0.79 |
| Vomiting | 288 (66.5) | 119 (71.3) | 407 (67.8) | 1.25 [0.85, 1.85] | 0.265 |
| Abdominal pain | 339 (78.3) | 139 (83.2) | 478 (79.7) | 1.38 [0.87, 2.22] | 0.179 |
| Diarrhea | 169 (39.0) | 80 (47.9) | 249 (41.5) | 1.44 [1.00, 2.06] | 0.049 |
| Bleeding | 137 (31.6) | 73 (43.7) | 210 (35.0) | 1.68 [1.16, 2.42] | 0.006 |
| Petechiaes | 223 (51.5) | 94 (56.3) | 317 (52.8) | 1.21 [0.85, 1.74] | 0.293 |
| Edema | 79 (18.2) | 59 (35.3) | 138 (23.0) | 2.45 [1.64, 3.65] | <0.001 |
| *Laboratory characteristics, median (IQR)* | | | | | |
| Leukocytes ×10³/μL | 4.5 (3.0 to 6.8) | 4.8 (3.5 to 7.9) | 4.6 (3.1 to 7.0) | 1.06 [1.01, 1.10] | 0.011 |
| Neutrophils ×10³/μL | 1.8 (1.1 to 3.1) | 1.9 (1.2 to 3.1) | 1.8 (1.1 to 3.1) | 1.05 [0.99, 1.11] | 0.085 |
| Lymphocytes ×10³/μL | 1.8 (0.9 to 3.1) | 2.1 (1.3 to 3.4) | 1.9 (1.0 to 3.2) | 1.08 [0.99, 1.19] | 0.091 |
| Platelets/μL | 90.0 (50.0 to 169.0) | 34.0 (23.0 to 73.0) | 71.5 (38.0 to 147.5) | 0.993 [0.991, 0.996] | <0.001 |
| ALT U/L | 54.0 (28.5 to 119.0) | 65.0 (40.0 to 126.0) | 58.0 (31.0 to 119.7) | 1 [0.99, 1.00] | 0.209 |
| No. | 419 | 167 | 586 | | |
| AST U/L | 101.5 (57.0 to 196.2) | 130.0 (80.0 to 234.5) | 112.0 (61.0 to 214.0) | 1 [1.000, 1.001] | 0.081 |
| No. | 420 | 167 | 587 | | |
| **IgG mg/dL n(%)** | | | | | |
| No. | 398 | 149 | 547 | | |
| Negative | 171 (43.0) | 43 (28.9) | 214 (39.1) | – | |
| Positive | 227 (57.0) | 106 (71.1) | 333 (60.9) | 1.86 [1.24, 2.81] | 0.003 |
| **Dengue serotype n(%)** | | | | | |
| Unknown | 194 (44.8) | 66 (39.5) | 260 (43.3) | 0.56 [0.38, 0.84] | 0.005 |
| DENV-1 | 42 (9.7) | 11 (6.6) | 53 (8.8) | 0.43 [0.20, 0.87] | 0.024 |
| DENV-2 | 121 (27.9) | 73 (43.7) | 194 (32.3) | Reference | |
| DENV-3 | 74 (17.1) | 17 (10.2) | 91 (15.2) | 0.38 [0.20, 0.68] | 0.002 |
| DENV-4 | 2 (0.5) | 0 (0.0) | 2 (0.3) | – | – |
| **Dengue lineages n(%)** | | | | | |
| 1V_D | 0 (0.0) | 1 (0.6) | 1 (0.2) | | |
| 1V_D.1 | 26 (6.0) | 7 (4.2) | 33 (5.5) | | |
| 1V_D.1.1 | 2 (0.5) | 0 (0.0) | 2 (0.3) | | |
| 1V_D.2 | 8 (1.8) | 2 (1.2) | 10 (1.7) | | |
| 1V_F | 1 (0.2) | 0 (0.0) | 1 (0.2) | | |
| 2II_F.1 | 1 (0.2) | 0 (0.0) | 1 (0.2) | | |
| 2II_F.1.1.2 | 71 (16.4) | 41 (24.6) | 112 (18.7) | | |
| 2II_F.1.1.5 | 2 (0.5) | 0 (0.0) | 2 (0.3) | | |
| 2III_D.2 | 36 (8.3) | 28 (16.8) | 64 (10.7) | | |
| 3III_B.3 | 1 (0.2) | 0 (0.0) | 1 (0.2) | | |
| 3III_B.3.2 | 0 (0.0) | 1 (0.6) | 1 (0.2) | | |
| 3III_C.1 | 71 (16.4) | 16 (9.6) | 87 (14.5) | | |
| 4II_B | 1 (0.2) | 0 (0.0) | 1 (0.2) | | |
| Unknown | 213 (49.2) | 71 (42.5) | 284 (47.3) | | |
| **Relevant dengue lineages n(%)** | | | | | |
| 1V_D.1 | 26 (6.0) | 7 (4.2) | 33 (5.5) | 0.82 [0.32, 1.88] | 0.66 |

*(Continued)*

**Table 1.** (Continued)

| Variable | Non-serious manifestation of dengue N = 433 | Serious manifestations of dengue N = 167 | Total N = 600 | OR [95 CI] | p- value |
|---|---|---|---|---|---|
| 2II_F.1.1.2 | 71 (16.4) | 41 (24.6) | 112 (18.7) | 1.76 [1.10, 2.80] | 0.017 |
| 2III_D.2 | 36 (8.3) | 28 (16.8) | 64 (10.7) | 2.37 [1.35, 4.15] | 0.002 |
| 3III_C.1 | 71 (16.4) | 16 (9.6) | 87 (14.5) | 0.69 [0.37, 1.23] | 0.223 |
| Other/unknown | 229 (52.9) | 75 (44.9) | 304 (50.7) | – | |

aLess than basic: No education or unfinished primary school. Basic/intermediate: Complete or incomplete high school. Advanced: Pursued further education after high school graduation (i.e., technical training, college or university). bBMI of children aged 0–18 years was classified based on standard deviations according to the WHO into underweight (below −2 SD), normal nutritional status (between −2 and +1 SD), overweight (between +1 and +2 SD) and obesity (above +2 SD). For adults (>18 years), BMI was classified using the following categories: underweight ($<18.5$ kg/m$^2$), normal (18.5 kg/m$^2$ to less than 25 kg/m$^2$), overweight (25 to less than 30 kg/m$^2$) and obesity (30 kg/m$^2$ or greater). No: number of observations with non-missing data

significant disease progression that do not strictly meet WHO severity criteria. Identifying risk factors for this broader category allows clinicians to detect not only rare severe cases but also a more frequently affected subgroup at high risk for complications.

While previous studies have identified age under 18 as a risk factor for severe dengue [26], our study did not find a significant association between age and patients with serious manifestations of dengue. However, our cohort's median age (13 years) was markedly younger than the statewide median age of patients with dengue (30 years). This finding may indicate that patients seeking care at a large referral hospital tend to be younger than those in the broader community, suggesting that adolescents with dengue may be at higher risk for hospitalization.

Chronic conditions such as diabetes, hypertension, obesity, and asthma have been linked to severe dengue, particularly in adults [10,26]. However, we did not observe a significant association in our primarily pediatric population. This aligns with prior research indicating that while comorbidities increase severity in adults [30,31], studies focusing on children often do not find a strong correlation [32].

Previous research has highlighted the role of viral factors in contributing to the severity of dengue infection. [25,33] Specific DENV serotypes have been associated with increased severity of clinical outcome. While a recent long-term cohort study from Nicaragua linked DENV-3 to severe primary infections, [33] our study identified DENV-2 as the serotype most strongly associated with serious manifestations. This finding aligns with studies from Paraguay [34] and Brazil, [35] highlighting the geographic and temporal variability of serotypes causing severe dengue infections and underscoring the need for tetravalent protection. Although many samples were collected late in the course of infection—leading to virus titers below the detection limits of our PCR and sequencing methods—we were still able to detect multiple lineages co-circulating during the study period. This finding helps explain the increased severity of the outbreak and underscores the need for genomic surveillance to monitor lineages with epidemiological and clinical importance. [18,36]

Previous infection with a heterologous DENV serotype has often been identified as a risk factor for severe dengue. [9] However, controversy exists due to a recent study that found severe disease to be equally frequent among primary and secondary infections in children, as classified based on IgM:IgG ratios using a dengue-specific ELISA, following WHO recommendations. [37]. In our cohort, prior dengue exposure was not independently associated with serious manifestations, and this finding aligns with those from the placebo arm of a multicenter phase 3 trial to assess the safety and efficacy of a dengue vaccine, in which the incidence of virologically confirmed dengue leading to hospitalization (as an indirect indicator of severe disease) was the same in seropositive and seronegative individuals (0.5 per 100 person-years [p-yrs] in participants with (101/21922 p-yrs) and without (41/8262 p-yrs) previous exposure). [38]

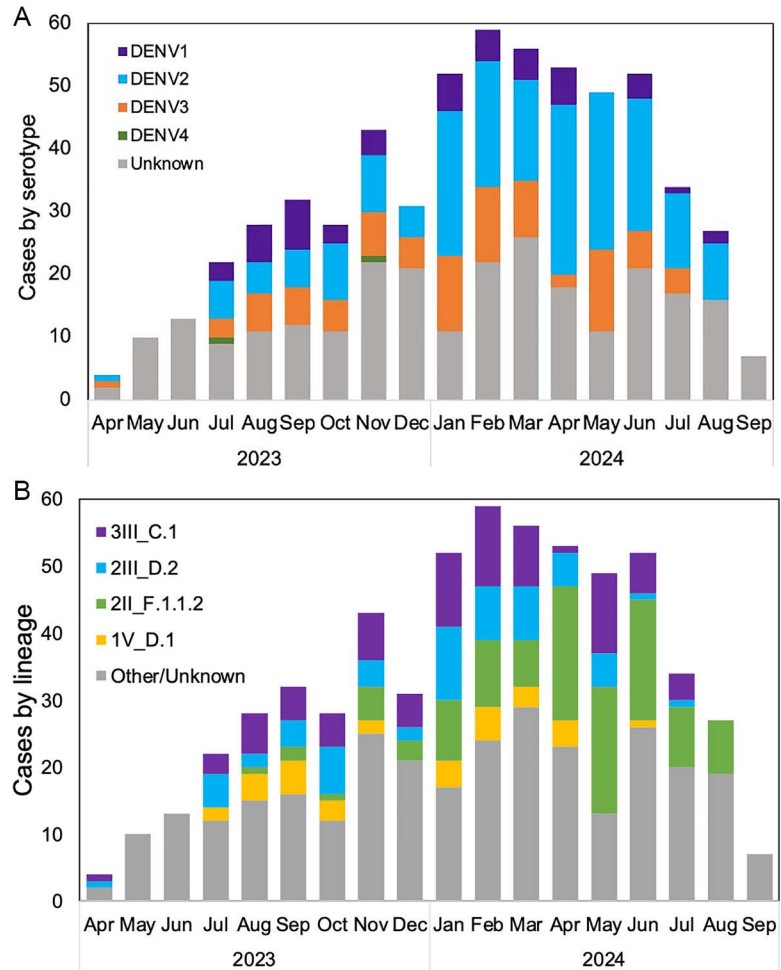

**Fig 3.** a. Evolution in the number and co-circulation of dengue virus serotypes by month from Apr 2023-Sep 2024. b. Evolution in the number and co-circulation of dengue virus lineages by month from Apr 2023-Sep 2024.

While in a large Nicaraguan cohort study, [39] the prevalence of severe disease in secondary vs. primary infections varied according to the infecting serotype (higher for DENV 2 and 4, similar for DENV 1 and 3), the effect of serostatus on the severity of illness according to infecting serotype was not observed in our cohort. These results highlight the complex interplay between infection severity, infecting serotype, and serotype-specific immunity, emphasizing the need for a deeper understanding of how the timing and sequence of infections influences transmission intensity and disease severity.

## Limitations

Several limitations should be considered. First, although living outside of Cali was associated with a higher risk of serious dengue manifestations, this may reflect referral bias rather than a true biological effect. Patients living outside of city limits may only be referred to the study hospital when illness is more advanced. Thus, the observed association may partly result from the data generation mechanism—namely that only more severe cases from outside Cali entered our sample— limiting the generalizability of this prognostic factor.

Second, while DENV-2 was identified as a risk factor for more severe disease, we were not able to identify the infecting serotype in a large proportion of samples. However, it is biologically unlikely that this technical limitation led to differential

**Table 2. Multivariable logistic regression modeling of demographic, clinical and viral characteristics independently associated with serious manifestations among patients with dengue[a].**

| Variable | OR | [95CI] | p-value |
|---|---|---|---|
| **Demographic Characteristics** | | | |
| **Age** | 1.00 | [0.98; 1.02] | 0.867 |
| **Sex** | | | |
| Female | – | – | – |
| Male | 1.18 | [0.74; 1.88] | 0.492 |
| **Level of education[b]** | | | |
| Less than basic | – | – | – |
| Basic/intermediate | 1.00 | [0.60; 1.68] | 0.988 |
| Advanced | 1.15 | [0.50; 2.58] | 0.742 |
| **City of origin** | | | |
| Cali | – | – | – |
| Other | 1.65 | [1.05; 2.58] | 0.029 |
| **Occupation** | | | |
| House wive | – | – | – |
| Employee | 0.39 | [0.13; 1.14] | 0.086 |
| Freelance work | 0.87 | [0.22; 3.45] | 0.839 |
| Unemployed/none | 0.89 | [0.31; 2.59] | 0.837 |
| Student | 0.46 | [0.17; 1.26] | 0.126 |
| **Dengue serotype** | | | |
| Unknown | 0.53 | [0.32; 0.87] | 0.01 |
| DENV-1 | 0.56 | [0.24; 1.24] | 0.16 |
| DENV-2 | – | – | – |
| DENV-3 | 0.38 | [0.18; 0.77] | 0.01 |
| **IgG** mg/dL | | | |
| Negative | – | – | – |
| Positive | 1.45 | [0.89; 2.39] | 0.134 |
| *Laboratory characteristics* | | | |
| Leukocytes ×10³/µL | 1.21 | [1.10; 1.33] | <0.001 |
| Lymphocytes ×10³/µL | 0.83 | [0.69; 0.98] | 0.037 |
| Platelets/µL | 0.993 | [0.989; 0.996] | <0.001 |
| AST U/L | 1.00 | [0.99; 1.001] | 0.867 |
| *Clinical symptoms* | | | |
| Mialgias/Arthralgias | 1.10 | [0.51; 2.53] | 0.811 |
| Abdominal pain | 0.98 | [0.56; 1.74] | 0.957 |
| Diarrhea | 1.24 | [0.80; 1.91] | 0.326 |
| Bleeding | 1.52 | [0.97; 2.40] | 0.067 |
| Edema | 2.15 | [1.32; 3.50] | 0.002 |

[a]A total of 533 observations with complete data were included in the model, of which 134 presented serious manifestations. [b]Less than basic: No education or unfinished primary school. Basic/intermediate: Complete or incomplete high school. Advanced: Pursued further education after high school graduation (i.e., technical training, college or university).

**Table 3. Outcomes of patients with dengue infection according to disease manifestations.**

| Variable | Non-serious manifestations of dengue N = 433 | Serious manifestations of dengue N = 167 | Total N = 600 | p- value |
|---|---|---|---|---|
| Number of days with fever, median (IQR) | 4 (3 –5) | 3 (3 –5) | 4 (3 –5) | 0.253 |
| ICU requirement, n (%) | 40 (9.2) | 64 (38.3) | 104 (17.3) | <0.001 |
| Number of days in the ICU, median (IQR) | 3 (2 –4) | 3 (2 –5) | 3 (2 –4) | 0.283 |
| Need for vasopressors, n (%) | 0 (0.0) | 11 (6.6) | 11 (1.8) | <0.001 |
| Death, n (%) | 2 (0.5) | 2 (1.2) | 4 (0.7) | – |

misclassification with respect to severity, and thus it is unlikely to substantially bias the identification of DENV-2 as a poor prognostic factor in this cohort.

Third, while we assessed a comprehensive set of clinical, laboratory, and virological factors, we did not analyze bio-markers such as IL-10, chymase, kynurenine, or transcriptomic signatures that have been identified as indicators for severe dengue. [10] Fourth, our assessment of prior DENV exposure relied on the VIDAS IgG assay; more precise methods, such as the EDIII-MMBA, could provide a deeper understanding of the effects of infection sequence, or varied levels of preexisting DENV specific antibodies on disease severity. Additionally, we did not evaluate the interval between successive infections, a factor hypothesized to influence outcomes. [36] Finally, while this study identified significant risk factors using a refined disease classification, future research should aim to develop and validate a predictive model that integrates demographic, clinical, virological, and immunological factors for improved severity assessment.

## Conclusion

The unpredictable nature of severe dengue, particularly given the late onset of clinical warning signs, underscores the need for robust risk identification. In this cohort, examined during the worst dengue outbreak in the Americas, certain clinical manifestations and laboratory findings were associated with increased severity and should be recognized early in the disease course to identify patients at risk not only for severe dengue but also for dengue with warning signs who go on to develop serious manifestations. Residing outside city limits increased the risk of serious manifestations, likely due to limited access to advanced, specialized care early in the disease course. DENV-2 infections heightened the risk of serious manifestations in this cohort, however, studies performed in different years and geographies have identified other sero-types as causative of severe dengue. In addition, the absence of a detectable effect of serostatus on the risk of serious outcomes across the different serotypes, underscores the necessity for tetravalent vaccines or other preventive meas-ures to protect the broad and growing populations at risk of dengue—regardless of prior dengue exposure. Importantly, the severity of dengue infection is influenced by a complex interplay between viral genotype, host immune history, and intrinsic host factors. These interactions contribute to variability in disease outcomes, highlighting the multifaceted nature of dengue pathogenesis. A comprehensive understanding of these interactions is essential to develop tailored strategies to prevent disease progression and ultimately reduce the global burden of dengue.

## Supporting information

**S1 Table. Sociodemographic and clinical characteristics of patients with Non severe and severe dengue virus infection according to WHO classification.**
(DOCX)

**S1 Data. Data set—Epidemiology and severity risk factors of dengue virus infection during the 2023–2024 out-break in Colombia.**
(XLSX)

## Author contributions

**Conceptualization:** Pio López, Diana M. Dávalos, Eduardo López-Medina.

**Data curation:** Daniela Torres-Hernández, Mara Gómez-Zambrano, Diana M. Dávalos, Eduardo López-Medina.

**Formal analysis:** Nathan D. Grubaugh, Erika Cantor, Eduardo López-Medina.

**Funding acquisition:** Nathan D. Grubaugh, Eduardo López-Medina.

**Investigation:** Daniela Torres-Hernández, Mónica A. Murillo-Ortiz, Isabel C. Hurtado, Mara Gómez-Zambrano, Eduardo López-Medina.

**Methodology:** Erika Cantor, Eduardo López-Medina.

**Project administration:** Daniela Torres-Hernández, Diana M. Dávalos.

**Resources:** Nathan D. Grubaugh, Verity Hill, Mallery I. Breban, Diana M. Dávalos, Eduardo López-Medina.

**Supervision:** Eduardo López-Medina.

**Visualization:** Eduardo López-Medina.

**Writing – original draft:** Daniela Torres-Hernández, Diana M. Dávalos, Eduardo López-Medina.

**Writing – review & editing:** Daniela Torres-Hernández, Nathan D. Grubaugh, Mónica A. Murillo-Ortiz, Isabel C. Hurtado, Verity Hill, Mallery I. Breban, Mara Gómez-Zambrano, Pio López, Erika Cantor, Diana M. Dávalos, Eduardo López-Medina.

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
