## [Decision Letter · Decision Letter 0]

24 Aug 2025

Epidemiology and Severity Risk Factors of Dengue Virus Infection During the 2023-2024 Outbreak in Colombia

Dear Dr. López-Medina,

Thank you for submitting your manuscript to PLOS Neglected Tropical Diseases. After careful consideration, we feel that it has merit but does not fully meet PLOS Neglected Tropical Diseases's publication criteria as it currently stands. Therefore, we invite you to submit a revised version of the manuscript that addresses the points raised during the review process.

Please submit your revised manuscript within 60 days Oct 23 2025 11:59PM. If you will need more time than this to complete your revisions, please reply to this message or contact the journal office at plosntds@plos.org. Please include the following items when submitting your revised manuscript:

We look forward to receiving your revised manuscript.

Kind regards,

Viviane Sampaio Boaventura, M.D.; MsC; Ph.D

Academic Editor

David Safronetz

Section Editor

Shaden Kamhawi

co-Editor-in-Chief

Paul Brindley

co-Editor-in-Chief

**Additional Editor Comments :**

This decision will be made after evaluating your responses and the corresponding revisions. In particular, Reviewers 1 and 2 raised concerns regarding the statistical analyses, and acceptance of the manuscript will be contingent upon addressing these issues.

**Journal Requirements:**

At this stage, the following Authors/Authors require contributions: Daniela Torres-Hernández, Nathan D. Grubaugh, Mónica A. Murillo-Ortiz, Isabel C. Hurtado, Verity Hill, Mallery I. Breban, Mara Gómez-Zambrano, Pio López, Erika Cantor, Diana M. Dávalos, and Eduardo López-Medina. Please ensure that the full contributions of each author are acknowledged in the "Add/Edit/Remove Authors" section of our submission form.

- ® on pages: 7, and 8.

1) If the funders had no role in your study, please state: "The funders had no role in study design, data collection and analysis, decision to publish, or preparation of the manuscript."

2) If any authors received a salary from any of your funders, please state which authors and which funders.

6) Please ensure that the funders and grant numbers match between the Financial Disclosure field and the Funding Information tab in your submission form. Note that the funders must be provided in the same order in both places as well. Currently, this grant (no. 1S10OD028669-01) is missing from the Funding Information tab.

**Reviewers' Comments:**

Reviewer's Responses to Questions

**Key Review Criteria Required for Acceptance?**

**Methods**

-Are the objectives of the study clearly articulated with a clear testable hypothesis stated?

-Is the study design appropriate to address the stated objectives?

-Is the population clearly described and appropriate for the hypothesis being tested?

-Is the sample size sufficient to ensure adequate power to address the hypothesis being tested?

-Were correct statistical analysis used to support conclusions?

-Are there concerns about ethical or regulatory requirements being met?

Reviewer #1: It is an interesting paper evaluating dengue cases in Cali, providing rich information about signs and symptoms. Although the findings are not novel, it serves as useful additional evidence. My main concerns regarding the paper relate to the statistical approach and the discussion of the findings.

Main:

Univariable selection (using variables with p < 0.05) is not a recommended approach for variable selection, as it often misses important variables in the model.

(https://pubmed.ncbi.nlm.nih.gov/8699212/ \ https://pubmed.ncbi.nlm.nih.gov/27896874/ \ https://onlinelibrary.wiley.com/doi/full/10.1002/bimj.201700067 ). Using background knowledge typically yields more stable and useful models. Alternatively, backwards elimination can be performed.

The decision to reuse the data for two different types of models is also not appropriate, as the model with signs/symptoms will lack all the important demographic variables, such as age.

This strategy of univariable selection can also generate spurious associations; for example, the association between education level and serious dengue. Considering the age group (median 13, IQR 8-19) evaluated, education level will be highly collinear with age, so individuals with an “Advanced” education level (I’m considering that it is a college degree, as there is no description in the methods section) will be at least around 23 years old. Using the data provided in the submission, individuals with “Advanced” education have a median age of 30 years (20 to 40), while those with basic-intermediate education have a median age of 16 (13 to 20). Any discussion related to this should regard age difference as the main mechanism, contrary to the hygiene hypothesis, which is discussed. Considering that both education level and occupation are highly correlated with age (within the age group of the study participants), a better variable to adjust for would be the socioeconomic status (income) of the household. Another variable with a similar issue is the city of residence. There is also a strong effect of city of residence, with areas outside Cali being prognostic of serious dengue. If more severe cases are referred to the hospital, this variable merely reflects the data generation mechanism, rather than any biological effect of living outside Cali.

Reviewer #2: 1. Which gene is used for dengue virus genotyping?

**Results**

-Does the analysis presented match the analysis plan?

-Are the results clearly and completely presented?

-Are the figures (Tables, Images) of sufficient quality for clarity?

Reviewer #1: Table 1 should also include a breakdown of comorbidities, even if they are not referred to as individual terms due to sample size constraints, enabling the reader to assess which types of comorbidities are being evaluated.

Table 1 should clearly indicate the presence of missing data (for example, IgG only has data for 547 patients)

It is also unclear what the interaction model is. (What are the reference levels in Table 3?) Considering the model:

Serious Dengue ~ Current infection * previous infection, it should present only one p-value evaluating the joint effect (Wald-test / likelihood test) instead of the individual p-values of the interactions (https://stats.oarc.ucla.edu/other/mult-pkg/faq/general/faqhow-are-the-likelihood-ratio-wald-and-lagrange-multiplier-score-tests-different-andor-similar/).

Moreover, the current p-values in Table 3 are also unclear, as there is significant overlap between the CIs for Dengue-2 despite p=0.05.

Additionally, it is uncertain whether the interaction model incorporates all sociodemographic characteristics or only the current and previous infections.

Reviewer #2: 1. The OR merely reflects the strength of association between exposure and outcome, but does not in itself establish a “high-risk factor.” ORs between 2–3 typically represent moderate risk and are common in clinical studies. Therefore, while risk factors such as residence outside Cali, DENV-2 infection, edema, bleeding, and thrombocytopenia appear statistically reliable, the interaction term between IgG status and DENV-2 infection remains inconclusive and requires further validation.

2. The multivariable model relies on variables selected solely based on univariate P values, which increases the risk of overfitting. It would be more appropriate to use clinically justified variables, along with diagnostics for collinearity (e.g., VIF) and model fit indices (e.g., AIC/BIC). As such, I consider the conclusions of this study somewhat overstated.

Reviewer #3: 1. Figure 1 - Where the data were extracted from for the period 2021-2024. The methodology and results were unclear.

2. Figure 1 - How the 400% value was reached was unclear.

3. Table 2a - I suggest including all serotypes in the regression analysis, not just DENV-2.

4. Lines 311-313. This refers to the group of IgG-positive patients. Is this correct? Make this clear in the text.

5. Table 3. Perform the analysis separately for the full cohort and the subcohort of seropositive patients. Include a supplementary table with the subcohort of seropositive patients.

6. Table 1 - What statistical test was used to generate the OR?

7. Table 1 - Present the limitation of unidentified serotypes in each group. Discuss this limitation as well.

8. Table 2a - Include other serotypes in the analysis as well, not just DENV-2.

9. Table 1 - There is a difference in the city between the severe and non-severe groups. Perform the analysis with the variables separately for the city of Cali and other cities, as this may be a confounding factor.

10. Lines 393-402. Cites several studies, but references only one. Further discuss the relationship between education and disease severity based on scientific findings.

11. Discuss why DENV2 was not associated with dengue severity in the analysis performed according to WHO stratification.

**Conclusions**

-Are the conclusions supported by the data presented?

-Are the limitations of analysis clearly described?

-Do the authors discuss how these data can be helpful to advance our understanding of the topic under study?

-Is public health relevance addressed?

Reviewer #1: The discussion should acknowledge the existence of studies exploring the same research question in other countries. Furthermore, any discussion regarding the different effects of serotype in the current study should recognise that nearly half of the cases lacked a defined serotype.

Reviewer #2: The authors’ statement that "a vaccine is needed that is safe and effective regardless of prior dengue infection" seems overly ambitious. In the study, participants identified as having "prior dengue infection (IgG positive)"—but how can one distinguish whether the IgG positivity resulted from prior natural infection or from previous vaccination?

**Summary and General Comments**

Reviewer #1: Minors:

Abstract

Report Confidence intervals together with OR, not p-values

Introduction.

The first paragraph is vague. “Models indicate that case numbers will continue to increase in the coming years unless effective measures are widely implemented”

Which type of model? Increase everywhere or only in Latin America? Etc, provide more information

There are some studies evaluating dengue with/without warning signs. example:

“Gonçalves, Bianca De Santis, et al. "Factors predicting the severity of dengue in patients with warning signs in Rio de Janeiro, Brazil (1986–2012)." Transactions of The Royal Society of Tropical Medicine and Hygiene 113.11 (2019): 670-677.”

Pinto, Rosemary Costa, et al. "Mortality predictors in patients with severe dengue in the State of Amazonas, Brazil." PloS one 11.8 (2016): e0161884.

It should be rephrased and take into account previous studies in the introduction (third paragraph)

Reviewer #2: This study is based on data collected during the large-scale dengue outbreak at HUV Hospital in Colombia (2023–2024), aiming to identify risk factors associated with "serious manifestations" of dengue. I believe the article has several issues in terms of methodological design, statistical modeling, and interpretation of results.

Reviewer #3: The causality of DENV2 with disease severity should be further discussed and explored if this is the focus of the study.

PLOS authors have the option to publish the peer review history of their article (what does this mean? ). If published, this will include your full peer review and any attached files.

**Do you want your identity to be public for this peer review?** For information about this choice, including consent withdrawal, please see our Privacy Policy .

Reviewer #1: **Yes: ** Thiago Cerqueira Silva

Reviewer #2: No

Reviewer #3: No

**Figure resubmission:**
---

## [Decision Letter · Decision Letter 1]

10 Nov 2025

Before final approval, however, a few **minor corrections** are still required to ensure clarity and consistency throughout the text. Once these are addressed, the manuscript will be ready for acceptance.

We look forward to receiving the revised version soon.

Revised Manuscript with Track Changes
Manuscript

Shaden Kamhawi

co-Editor-in-Chief

Paul Brindley

co-Editor-in-Chief

**Additional Editor Comments:**

Additionally, in line 80, the sentence “whether or not they have had exposed to dengue in the past” should be revised to clarify the limitation of relying on registry data. We suggest rephrasing it to:

**Reviewers' comments:**

**Key Review Criteria Required for Acceptance?**

**Methods**

-Are the objectives of the study clearly articulated with a clear testable hypothesis stated?

-Is the study design appropriate to address the stated objectives?

-Is the population clearly described and appropriate for the hypothesis being tested?

-Is the sample size sufficient to ensure adequate power to address the hypothesis being tested?

-Were correct statistical analysis used to support conclusions?

-Are there concerns about ethical or regulatory requirements being met?

Reviewer #1: (No Response)

Reviewer #2: (No Response)

Reviewer #3: The authors reformulated and better described some statistical analyses in the manuscript. I suggest verifying that this covers all the statistical concerns raised.

**Results**

-Does the analysis presented match the analysis plan?

-Are the results clearly and completely presented?

-Are the figures (Tables, Images) of sufficient quality for clarity?

Reviewer #1: Minor comments on the Table 1/2 lack the measurement unit of the laboratory characteristics. Also, the regression with laboratory characteristics as a continuous variable is not informative, as there is a range of values normal for each characteristic. A better approach would be to characterise as below/normal/above the reference values for each characteristic

Reviewer #2: (No Response)

Reviewer #3: The results were better described in this version of the manuscript, although they still reinforce the causal relationship between severe dengue and DENV-2.

**Conclusions**

-Are the conclusions supported by the data presented?

-Are the limitations of analysis clearly described?

-Do the authors discuss how these data can be helpful to advance our understanding of the topic under study?

-Is public health relevance addressed?

Reviewer #1: (No Response)

Reviewer #2: (No Response)

Reviewer #3: The authors better described the limitations of the study in this new version. However, the conclusions regarding causality could be more nuanced.

**Editorial and Data Presentation Modifications?**

Reviewer #1: (No Response)

Reviewer #2: (No Response)

Reviewer #3: (No Response)

**Summary and General Comments**

Reviewer #1: (No Response)

Reviewer #2: (No Response)

Reviewer #3: The current version of the manuscript has incorporated improvements in the description of the methodology, results, discussions and limitations.

PLOS authors have the option to publish the peer review history of their article (what does this mean? ). If published, this will include your full peer review and any attached files.

**Do you want your identity to be public for this peer review?** For information about this choice, including consent withdrawal, please see our Privacy Policy .

Reviewer #1: **Yes: ** Thiago Cerqueira-Silva

Reviewer #2: No

Reviewer #3: No

**Figure resubmission:**

**Reproducibility:** To enhance the reproducibility of your results, we recommend that authors of applicable studies deposit laboratory protocols in protocols.io, where a protocol can be assigned its own identifier (DOI) such that it can be cited independently in the future. Additionally, PLOS ONE offers an option to publish peer-reviewed clinical study protocols. Read more information on sharing protocols at https://plos.org/protocols?utm_medium=editorial-email&utm_source=authorletters&utm_campaign=protocols

---

## [Editor Report · Decision Letter 2]

27 Nov 2025

Dear Dr. López-Medina,

We are pleased to inform you that your manuscript 'Epidemiology and Severity Risk Factors of Dengue Virus Infection During the 2023-2024 Outbreak in Colombia' has been provisionally accepted for publication in PLOS Neglected Tropical Diseases.

Best regards,

Viviane Sampaio Boaventura, M.D.; MsC; Ph.D

Academic Editor

David Safronetz

Section Editor

Shaden Kamhawi

co-Editor-in-Chief

Paul Brindley

co-Editor-in-Chief

---

## [Editor Report · Acceptance letter]

Dear Dr. López-Medina,

We are delighted to inform you that your manuscript, "Epidemiology and Severity Risk Factors of Dengue Virus Infection During the 2023-2024 Outbreak in Colombia," has been formally accepted for publication in PLOS Neglected Tropical Diseases.

Best regards,

Shaden Kamhawi

co-Editor-in-Chief

Paul Brindley

co-Editor-in-Chief
